# Autotrophic Acetate Production under Hydrogenophilic and Bioelectrochemical Conditions with a Thermally Treated Mixed Culture

**DOI:** 10.3390/membranes12020126

**Published:** 2022-01-21

**Authors:** Lorenzo Cristiani, Jacopo Ferretti, Mauro Majone, Marianna Villano, Marco Zeppilli

**Affiliations:** Department of Chemistry, Sapienza University of Rome, P.le Aldo Moro 5, 00185 Rome, Italy; lorenzo.cristiani@uniroma1.it (L.C.); jacopo.ferretti@uniroma1.it (J.F.); mauro.majone@uniroma1.it (M.M.); marianna.villano@uniroma1.it (M.V.)

**Keywords:** bioelectrosynthesis, bioelectrochemical system, homoacetogenesis, methanogenesis inhibition, CO_2_ bioreduction

## Abstract

Bioelectrochemical systems are emerging technologies for the reduction in CO_2_ in fuels and chemicals, in which anaerobic chemoautotrophic microorganisms such as methanogens and acetogens are typically used as biocatalysts. The anaerobic digestion digestate represents an abundant source of methanogens and acetogens microorganisms. In a mixed culture environment, methanogen’s inhibition is necessary to avoid acetate consumption by the presence of acetoclastic methanogens. In this study, a methanogenesis inhibition approach based on the thermal treatment of mixed cultures was adopted and evaluated in terms of acetate production under different tests consisting of hydrogenophilic and bioelectrochemical experiments. Batch experiments were carried out under hydrogenophilic and bioelectrochemical conditions, demonstrating the effectiveness of the thermal treatment and showing a 30 times higher acetate production with respect to the raw anaerobic digestate. Moreover, a continuous flow bioelectrochemical reactor equipped with an anion exchange membrane (AEM) successfully overcomes the methanogens reactivation, allowing for a continuous acetate production. The AEM membrane guaranteed the migration of the acetate from the biological compartment and its concentration in the abiotic chamber avoiding its consumption by acetoclastic methanogenesis. The system allowed an acetate concentration of 1745 ± 30 mg/L in the abiotic chamber, nearly five times the concentration measured in the cathodic chamber.

## 1. Introduction

The increase in CO_2_ generation is bound to further increase as a result of the economic growth and urbanization [1]. As CO_2_ is the most abundantly emitted greenhouse gas responsible for most of the radiative forcing of global warming [2,3]. CO_2_ removal and recycle are a major challenge to reduce emissions by 2030 as a global collective goal to limit the global warming to an increase of 1.5 °C [4,5]. In addition to substitution of fossil fuels with renewable energies, several capture and storage technologies (CCS) such as adsorption, membrane separation, and geological storage are being developed to mitigate CO_2_ emissions [6]. On the other hand, technologies that allowed the CO_2_ capture and conversion into valuable products has been named carbon capture and utilization chain (CCU) [2]. In this context, biological CCU approaches, which allows for the CO_2_ fixation in liquid [7], solid [8], gaseous [9] fuels represent an effective strategy towards a sustainable zero-emission economy. 

Among the biological CCU approaches, the fixation of CO_2_ into acetate performed by acetogenic microorganisms, also named homoacetogens (or acetogens), results particularly advantageous due to the possibility to perform the conversion of CO_2_ into acetate, i.e., it allows the conversion of highly oxidized inorganic carbon into an organic molecule. Acetogens microorganisms are chemoautotrophic strict anaerobes capable of growing on gaseous mixtures, which produce acetate from CO_2_ utilizing H_2_ as electron donor through the reductive pathway of acetyl coA (or Wood-Ljungdahl) [10]. An innovative approach for the reducing power supply to chemoautotrophic microorganisms is represented by the use of bioelectrochemical systems which consist in the utilization of an electrochemical device for the stimulation of the microbial metabolism though the interaction with polarized electrodes. Microbial electrosynthesis (MES) is a particular application of the bioelectrochemical systems in which by means of a cathode-driven process the direct absorption of hydrogen (mediated hydrogen transfer) or electron (extracellular electrons transfer EET) by the biofilm [11,12,13] occurs to catalyze the reduction in carbon dioxide and the extracellular generation of valuable reduced multi-carbon products. 

The use of mixed cultures is attractive due to their resilience to stress and fluctuations [12] and, especially, for its large availability. In fact, acetogens microorganisms are abundant in anaerobic environments like in anaerobic digestate [13]; however, methanogens, which are also present in anaerobic digestate, represent a bottleneck due to the acetoclastic acetate degradation [14] and the hydrogenophilic methanogenesis [15] which compete with acetogens for the reducing power [16]. Therefore, the selection of an appropriate acetogenic inoculum requires the methanogenesis inhibition to avoid product depletion and substrate competition. Methanogenesis inhibition on laboratory scale is usually obtained by 2-bromo-ethane sulfonate [17] or chloroform [18]. However, other inhibition strategies are being studied with a view to sustainable scalability process, such as pH control [19], ultrasonication pretreatment [20], aeration [21] and heat pretreatment [22]. The latter currently offers the best compromise for the selection of the acetogenic metabolism in mixed culture because some acetogens bacteria, following the thermal shock form endospores, while the non-spore-forming methanogens respond to the thermal shock with a long-term inhibition [22]. The effectiveness of thermal shock inhibition has already been confirmed in previous works showing the complete methanogenesis inhibition for a period of about 15–20 days [23]. 

In this study, the thermal treatment procedure was applied to an anerobic digestate coming from a full scale process to select an acetogenic inoculum, moreover, a continuous flow microbial electrolysis cell equipped with an anion exchange membrane (AEM) has been set up to overcome the limitation observed under batch mode. The transport of acetate across an AEM has been proposed [24] and demonstrated in recent studies [25]. In fact, removing the acetate from the cathode compartment and concentrating it in an abiotic compartment represents an effective solution to the coexistence between producers and competitors’ microorganisms. Furthermore, in this way the simultaneous production and extraction of the acetate in a combined device is achieved according to the recent “In situ product recovery” consisting in the integration of the biological production and product extraction the same reactor [26].

## 2. Materials and Methods

### 2.1. Inoculum’s Pretreatment

An anerobic digestate coming from a full scale aerobic digestor located in Treviso (Italy), was used as acetogens source. The anaerobic digestate was washed in the mineral medium having the following composition for 1.0 L of distilled water: 4.0 g K_2_HPO_4_, 0.125 g NH_4_Cl, 0.05 g CaCl_2_ 2 H_2_O, 0.1 g MgCl_2_ 6H_2_O, 1.0 mL of vitamin solution and 10 mL of metal solution reported in previous studies [27]. The anerobic digestate was transferred to a crystallizer and left to dry in an electric muffle furnace at 60 °C for a few days. Afterwards, the dried sludge was powdered with a mortar and sieved to select particles smaller than 500 µm. The resulting powder was transferred to a ceramic pot and treated in an electric muffle furnace at 120 °C for two hours. Subsequently, the powder was resuspended in the mineral medium and, to ensure the absence of organic substrates produced by the thermal treatment, it was washed 3 times with the same medium. 

### 2.2. Hydrogenophilic and Bioelectrochemical Batch Experiments

Hydrogenophilic tests were conducted at a pH value of 7.5 by adding molecular hydrogen to the gaseous phase in a 245 mL serum bottle, containing 150 mL of inoculum (raw sludge or heat-treated sludge with TSS of 2 g/L). An endogenous control test was associated with each test, i.e., a replica without H_2_, in order to determine the net production of acetate caused by the presence of trace of organic material. Hydrogen was added to the serum bottles whenever it was completely depleted while CO_2_ was present in large excess as bicarbonate and CO_2_ in the liquid and gaseous phase, respectively. 

The bioelectrochemical experiment was conducted using a H-Cell reactor which consisted of two 150 mL borosilicate glass bottles connected by their side flange and separated by an 0.01 m^2^ anion exchange membrane (AEM; FUMASEP^®^ FAS, FumatechGmbH, Bietigheim-Bissingen, Germany). Both anode and cathode electrodic material consisted of graphite rods. The bioelectrochemical cell was operated with a three-electrode configuration with the cathode as working electrode and the anode as counter electrode; an Ag/AgCl, placed near the cathode, was the reference electrode. The potential was controlled by a potentiostat (IVIUM-N-STAT) at a potential of −0.9 V vs. SHE (standard hydrogen electrode). The cathodic compartment was filled with 100 mL of inoculum while the anodic chamber was abiotic and contained 100 mL of mineral medium. 

### 2.3. Continuous Flow Bioelectrochemical Reactor

The continuous BES (Figure 1) was composed of two identical 0.86 L plastic chambers, named anodic and cathodic chamber. The chambers were separated by an anion exchange membrane with a surface area of 0.289 m^2^. Two different anion exchange membranes (AEM) were tested in the continuous bioelectrochemical reactor, the FUMASEP^®^ FAD and the FUMASEP^®^ FAS membranes (Fumatech GmbH, Bietigheim-Bissingen, Germany), both AEM membranes were constituted by an aramidic structure functionalized with quaternary ammonium. The FAD membrane has a selectivity higher than 87% and a stability between pH 1 and 9, FAD membrane t is typically used in dialysis processes for the recovery of free acids. On the other hand, FAS membrane has a higher selectivity between 94 and 97%, with a larger pH stability in the range of pH 1–14, FAS membrane has a greater capacity of proton blocker, and it is used for desalination processes, concentration of salts, acids and bases, nitrogen removal from drinking water. The reactor was operated with a three-electrode configuration by controlling the cathodic potential at −0.65 V vs. SHE with an IVIUM-N-STAT potentiostat. The cathodic compartment was filled with granular graphite and was inoculated with 200 mg of the same pretreated and completely dry sludge used for the batch tests. The anodic chamber, on the other hand, contained a mixed metal oxide (MMO) electrode (Magneto, Netherland) inserted in a silica bed which constituted the inert material with a mechanical support function. Both the anodic and cathodic chambers were equipped with an internal recirculation pump for the liquid phase, to guarantee a condition of perfect mixing. The reactor was continuously fed with pure CO_2_, which was contained in a double valve plastic bag, constantly recirculated at a flow rate of 60 mL/min. Moreover, a daily refill with mineral medium was necessary in the cathodic chamber in order to counterbalance the electroosmotic diffusion of water from the cathode to the anode chamber.

### 2.4. Analytical Procedures

The detection of H_2_, CO_2_, CH_4_ was carried out with gas-chromatographic analysis of the headspace of the following: the H cells; the serum bottles; and the gaseous phase of the reactor using a Dani Master GC equipped with a packed column and TCD detector set with the following conditions: injection volume 50 µL, carrier gas nitrogen, flow 12 mL/min, inlet temperature 120 °C, column temperature 70 °C and TCD detector at 150 °C. To sample the gaseous phase, a Gas tight syringe (Hamilton Company, Reno, Nevada, USA) was used. The analysis of volatile fatty acids (VFA) was performed by injecting 1 µL of liquid sample into a Dani Master gas chromatograph equipped with an FID detector with the following configuration: helium as carrier gas, flow 25 mL/min, temperature injector 200 °C, column temperature 175 °C and FID detector at 200 °C. The liquid samples were filtered (cellulose acetate filters, diameter 25 mm, pore diameter 0.2 µm) and prepared by adding to 1.0 mL of liquid sample (diluted or undiluted as needed) 100 µL of 0.33 M oxalic acid. Inorganic carbon was measured by TOC (total carbon analyzer)-V CSN (Shimadzu) on filtered samples.

### 2.5. Data Elaboration

#### 2.5.1. Conversion of Concentration into Meq

Equivalents of acetate, methane, and hydrogen were obtained knowing stoichiometry of different reduction reactions (Equations (1)–(3)):2H^+^ + 2e^−^ → H_2_(1)
CO_2_ + 8e^−^ + 8H^+^ → CH_4_ + 2H_2_O(2)
2CO_2_ + 8e^−^ + 8H^+^ → CH_3_COOH + 2H_2_O(3)

The conversion of molar concentration into equivalent of electrons were obtained by Equation (4): meq = mM × ne^−^ × V(4)
where mM is the millimolar concentration, n is the stoichiometric coefficient of the electrons involved in the respective reduction reaction and V is the volume of liquid or gas phase of the experiment. 

#### 2.5.2. Methane and Acetate Production Rate (rCH_4_ and rCH_3_COOH)

In the batch tests the production rate of methane and acetate was evaluated using Equation (5):(5)$$rCH_{4}\;or\;rCH_{3}COOH\left(\frac{meq}{Ld}\right)=\frac{meq\left(CH_{4}/CH_{3}COOH\right)}{V\left(L\right)\times \Delta t\left(d\right)}$$

In which the cumulative production of methane or acetate was divided by the time of the experiment while V represents the liquid volume of the corresponding test. 

In the continuous bioelectrochemical reactor’s anodic chamber, not having acetate production, the production rate was equal to the product between the concentration of acetate in the anodic chamber and spill rate of liquid phase. To summarize, considering a steady state condition, the acetate production was calculated by Equation (6).
(6)$$r\left[CH_{3}COOH\right]=\frac{Q_{out\;anode}\times \left[CH_{3}COOH\right]}{V_{cathode\;\left(L\right)}}$$

#### 2.5.3. Cathodic Capture Efficiency (CCE%)

This parameter described in Equation (7) was calculated only for bioelectrochemical tests and represents the fraction of the electrons flowing in the external circuit effectively used for the reduction in carbon dioxide to methane or acetate or for the reduction in H^+^ to molecular hydrogen.
(7)$$\mathrm{CCE}\left(\%\right)=\frac{meq\left(CH_{4}/CH_{3}COOH\right)}{\int_{t=n}^{t=n+1}\frac{idt}{F}}\times 100$$

#### 2.5.4. Inorganic Carbon Balance in the Continuous Bioelectrochemical Reactor

The inorganic carbon mass balance was calculated in the continuous bioelectrochemical reactor to assess the different mechanisms involved in the CO_2_ removal from the influent cathodic gaseous stream. The CO_2_ removal in the cathodic chamber is expressed by the daily removal ΔCO_2_ (mmol/d), which correspond to the variation of CO_2_ concentration measured in the cathodic gaseous phase as reported in Equation (8).
(8)$$\Delta CO_{2}=\frac{\left(\left[CO_{2\;day\;1\;cathode}\right]-\left[CO_{2\;day2\;cathode}\right]\right)\times V_{total}\;}{\Delta t}$$

The cathode chamber inorganic carbon mass balance, described in Equation (9), takes into account the three different CO_2_ removal mechanisms which consist of acetate production (rCH_3_COO^−^), methane production (rCH_4_) and bicarbonate migration from the cathode to the anode (HCO_3_^−^ _transf_.). As previously reported [28], the bicarbonate formation was a direct consequence of the alkalinity generation in the cathodic chamber promoted by the migration of anions different from hydroxyls.
(9)$$\Delta CO_{2\;cat}=r\;CH_{4}+r\;CH_{3}COO^{-}+HCO_{3\;trasf.}^{-}$$

The inorganic carbon mass balance in the anode is described by Equation (10)
(10)$$HCO_{3\;trasf.}^{-}=Q_{out}\times \left[CO_{2\;out\;anode}\right]+Q_{spill}*\left[HCO_{3\;anode}^{-}\right]$$
in which the bicarbonate migration from the cathode to the anode represents the influent inorganic carbon in the anode chamber while the two outlets are the daily CO_2_ coming out from the anode headspace (Q_out_ gaseous flow rate outlet of anode, CO_2out_ anode molar concentration of CO_2_ in the gaseous outlet) while the other term represents the bicarbonate removed with the daily spill of anolyte (Q_spill_ outlet liquid flow rate from anode, HCO_3_^−^_anode_ molar concentration of HCO_3_^−^ in the anolyte). Therefore, the sum of the two partial mass balance equation described the global inorganic carbon mass balance, as described in Equation (11):(11)$$\Delta CO_{2}=rCH_{4}+rCH_{3}COO^{-}+Q_{out}*\left[CO_{2\;out\;anode}\right]+Q_{spill}\times \left[HCO_{3\;anode}^{-}\right]$$

## 3. Results

### 3.1. Hydrogenopilic Batch Tests

The hydrogenophilic tests were conducted to test the effectiveness of the thermal treatment of the anaerobic digestate as previously reported [23]. As expected, the net acetate production with the raw sludge resulted quite low, as shown by Figure 2A,C, which reported the hydrogenophilic and the control test, respectively. Indeed, in the hydrogenophilic raw digestate test, acetate reached a maximum net production of 0.92 meq on day 43, which corresponded to an acetate net production rate of 0.6 mg/Ld (0.09 meq/Ld). In the raw sludge test, methanogenesis represented the main autotrophic mechanisms that reached a maximum amount of 9.2 meq on the 29th day in the hydrogenophilic test (Figure 2A) and 8.7 meq on the 27th day in the control test (Figure 2C), indicating the predominance of acetoclastic methanogenesis, which probably utilized the residual organic matter present in the digestate. On the contrary, as previously reported [23], the hydrogenophilic test conducted with the thermal treated inoculum (Figure 2B) was able to produce only acetate till day 20, when also methane started to be detected in the serum bottle headspace. Consequently, we can affirm that the heat treatment inhibits the methanogens, obtaining an acetogenic inoculum, as reported in a previous microbial community characterization performed on similar matrix [23]. The maximum acetate concentration reached in the hydrogenophilic test with thermal treated inoculum was around 300 mg/L (corresponding to 6.0 meq of acetate), with a maximum net production rate of 11.0 mg/Ld (1.43 meq/Ld), obtained through the difference with the acetate product in the control test (Figure 2D). Despite the absence of hydrogen, the control test reported in Figure 2D, showed an acetate production of 2.8 meq, which corresponded to a production rate of 0.41 meq/Ld. Methanogenesis reactivation was observed in both hydrogenophilic and control tests at around day 20, with a net methane production rate of 0.1 meq/Ld. 

### 3.2. Bioelectrochemical Batch-Test

The batch bioelectrochemical tests were conducted only with the thermal treated sludge at a cathodic potential of −0.9 vs. SHE, due to the necessity of ensuring H_2_ production for the acetogenic reaction already described in the literature [23]. During the bioelectrochemical test the average current generated by the electrochemical system was −0.3 ± 0.1 mA (Appendix A), while the cell voltage between anode and cathode reached a value of 2.8 ± 0.8 V. Acetate was produced in the cathodic chamber (Figure 3A) and it also migrated to the abiotic anodic chamber through the membrane due to the migration mechanism promoted by the established electric field. During the operation of the bioelectrochemical cell, acetate concentration reached a cumulative production of 0.5 meq (Figure 3B) which corresponded to a concentration around 30 mg/L. Considering the total amount of acetate produced, showed in Figure 3B, an average acetate production rate of 0.2 meq/Ld was observed, which in turn corresponded to a cathode capture efficiency (CCE) of 17.0 ± 0.2%. Moreover, differently from the hydrogenophilic test and as also reported in previous experiments [23], under bioelectrochemical conditions the methanogenesis reactivation was considerably faster, i.e., as also showed in Figure 3B methane was detected starting from day 8. This fact underlines the possibility to have a direct electron transfer between electrodic surface and methanogens as firstly described in the literature [29]. Moreover, Figure 3B showed a decrease in acetate concentration as a consequence of the methanogenesis establishment in the cathodic compartment. In fact, the CCE for acetate on day 8 decreased to 13.0 ± 0.2%, while methanogenesis quickly reached a CCE of 20.0 ± 0.2% with a production rate of 2.2 meq/Ld. After the methanogenesis reactivation, a hydrogen decrease was observed in the H-cell headspace (Appendix A). 

### 3.3. Continuous Bioelechtrochemical Reactor

The test conducted with the continuous flow bioelectrochemical reactor inoculated with the thermally treated sludge was performed at a cathodic potential of −0.65 vs. SHE. The average current obtained at the applied potential resulted on average 44.0 ± 2.2 mA. As shown in Figure 4, a lag phase of approximately 20 days was observed in terms of acetate production, with a consequent high production of hydrogen (Appendix A). Despite the possibility to convert COD in other VFA anions by an electrofermentation pathway [30], no other volatile fatty acids other than acetate have been detected in all the reported bioelectrochemical tests. The system entered a steady state condition at around the 20th day of operation, with an acetate concentration of 102 ± 14 mg/L in the cathodic chamber while, thanks to the migration of the acetate, the anodic acetate concentration reached 526 ± 95 mg/L, which resulted in a five times higher acetate concentration. The daily acetate production rate was 0.7 ± 0.3 meq/L d (approximately 4.85 mg/Ld) with a current recovery into acetate (CCE) of 2.3 ± 2.0% (Table 1). During the first operating period, an increase in cell voltage from −2.5 to −12 V was observed (Appendix A). At day 34, the FAD membrane was replaced with a FAS membrane due to the presence of an insoluble precipitate on the FAD membrane surface (Appendix A). The insoluble precipitate was constituted by carbonate and copper, which was not quantified but it was determined by qualitative assay (Appendix A).

Starting this new operation period, named AEM-FAS period, an average current of 52.1 ± 2.5 mA was obtained. Moreover, as shown in Figure 4, a new steady state period in which the acetate in the anodic chamber reached a value of 200 meq (which corresponds to a concentration of 1745 mg/L) was observed. The migration of the acetate from the cathode to the anode, together with the migration of all the other anionic species, promoted the electroosmotic diffusion of liquid phase from the cathode to the anode. Indeed, it was necessary to reintroduce the liquid phase in the cathodic chamber and spill the liquid phase from the anodic compartment, which was performed at the same rate of 0.01 L/d. Considering the anodic spill as the daily outlet flow from the bioelectrochemical reactor, an acetate production rate of 2.4 ± 0.95 meq/Ld (approximately 22 mg/Ld) was obtained during the FAS operational period, which in turn corresponded to a higher CCE of 7.2 ± 2.1%. The methane production started from day 20, and it was constantly present with an average production rate of 0.3 ± 0.2 meq/Ld during the AEM-FAD operational period, while after the introduction of the AEM-FAS membrane a higher production rate of 1.2 ± 0.8 meq/Ld was observed. Moreover, the methanogenesis mechanisms allowed for the justification of the major part of the current, indeed, during the AEM-FAD period a CCE of 14.6 ± 2.0% was detected while in the AEM-FAS period, a CCE of 58.0 ± 3.0% was obtained. As also reported in Figure 4, the higher methane production rate was not dependent by the AEM membrane utilized, but it was probably correlated with the acclimatation of the methanogens community in the cathodic biofilm. Despite of the cathodic methane production, which consumed mostly of the produced current, the utilization of the continuous flow bioelectrochemical reactor allowed for the continuous extraction of the product from the cathodic chamber preventing its conversion into methane by the acetoclastic methanogens. Moreover, also during the AEM-FAS operational period, an increase in the cell voltage from −2.5 to −8.0 V was observed (Appendix A), probably indicating the formation of the insoluble copper-precipitate on the membrane surface.

Despite the accomplished acetate production and extraction, higher performances in terms of acetate production (0.7 g/Ld) and concentration (up to 35 g(/L) have been described in the literature thanks to the utilization of specific antibiotics in the extraction chamber (anode compartment) and by the selection of an enriched acetogenic inoculum [31]. 

### 3.4. Inorganic Carbon Mass Balance of the Continuous Flow Bioelectrochemical Reactor

The inorganic carbon mass balance was set to give a comprehensive description of the different mechanisms involved in CO_2_ removal and utilization. As reported in Table 1 and described in Figure 5, with the daily methane and acetate production rates a balance of inorganic carbon was made, the parameters of which are shown in Table 1, i.e., the carbon dioxide daily removal, the production rate of methane and acetate, the migration rate of bicarbonate. 

For the entire duration of the experiment with both membranes, the bicarbonate concentration in the cathodic compartment was higher than the concentration of bicarbonate in the anodic compartment (Figure 6). This happened because, the acid pH inside the anodic chamber (3.2 ± 0.5) promoted the bicarbonate decomposition into CO_2_, which was collected as gaseous effluent from the anodic chamber. During the AEM-FAD period, the production of acetate and methane contributed to the removal of CO_2_ by 1.7 ± 0.5% and 17.0 ± 5.6%, while, during AEM-FAS period, 2.7 ± 0.2% and 35.0 ± 5.0% of the CO_2_ were converted into acetate and methane, respectively. By applying the global equation for the inorganic carbon mass balance, a considerable amount of undetermined inorganic carbon was assessed, i.e., 75 ± 14% and the 63 ± 12% of the removed CO_2_ during the AEM-FAD and AEM-FAS operational period were not quantified. The latter result probably indicates an additional unknown CO_2_ removal mechanism which was identified in the precipitation of an insoluble copper-based carbonate precipitate present on the side of the AEM membranes. In fact, the presence of a copper was determined by a qualitative assay (Appendix A) [32]. Table 1 summarized the different identified CO_2_ removal mechanisms during the two operational periods of the continuous flow bioelectrochemical reactor. 

## 4. Conclusions

This study confirmed the effectiveness of the heat treatment on an anaerobic sludge to promote methanogenesis inhibition. In the thermally treated hydrogenophilic batch test, the complete methanogenesis inhibition was maintained for 20 days, allowing a higher production of acetate. Bioelectrochemical batch test carried out with the thermally treated sludge, showed a 2 times faster reactivation of methanogens than in the hydrogenophilic one, indicating a possible direct electron transfer mechanism between methanogens and electrodic material. In the bioelectrochemical reactor, despite the low acetate production efficiency (7.2%) and a high methane efficiency (58%), the acetate has been successfully concentrated inside the anodic chamber against the concentration gradient. This was observed by using an AEM-FAS membrane, which, having a greater specificity, probably allowed a faster migration. In fact, with the previous AEM-FAD period, the daily acetate production and the CCE efficiency were 0.7 ± 0.3 meq/Ld and 2.3 ± 2.0%, respectively. The carbon balances carried out in the continuous bioelectrochemical reactor showed that the main mechanism for removing CO_2_ was its absorption in the liquid phase as bicarbonate/carbonate with the subsequent formation of a copper-based precipitate in the anodic chamber, the origin of which requires more specific investigation. Despite the low acetate production efficiencies, this system has shown that, starting from waste sludge such as anaerobic digestate, it is possible to produce and preserve acetate by using an appropriate membrane even in presence of a competitive reaction such as methanogenesis.

## Figures and Tables

**Figure 1 membranes-12-00126-f001:**
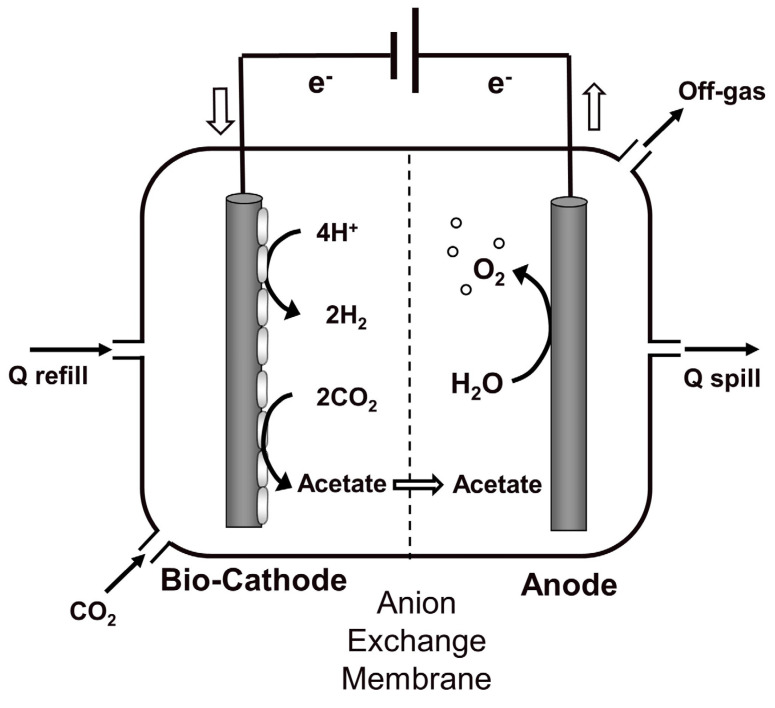
Scheme of the continuous flow bioelectrochemical reactor.

**Figure 2 membranes-12-00126-f002:**
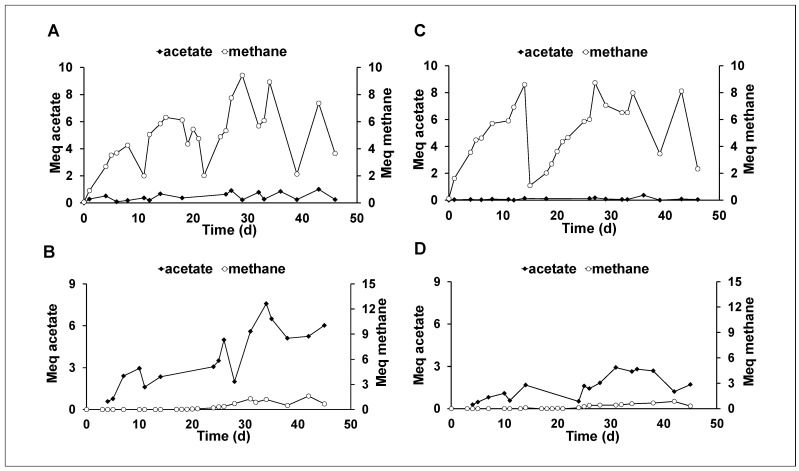
Profile of the amount of acetate and methane obtained in the hydrogenophilic experiments inoculated with raw sludge (**A**) and thermal treated sludge (**B**), with the respective endogenous controls (**C**,**D**).

**Figure 3 membranes-12-00126-f003:**
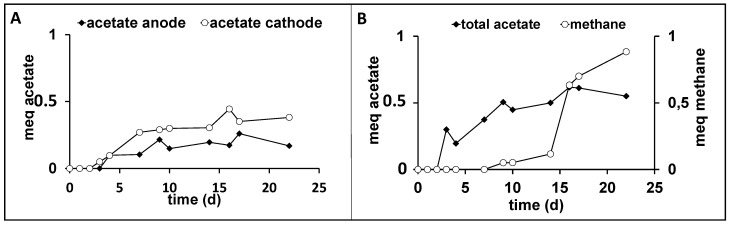
Time profile of the amount of acetate in the anodic and cathodic chamber (**A**), and of the total acetate and methane (**B**) obtained in the bioelectrochemical experiment inoculated with heat treated sludge.

**Figure 4 membranes-12-00126-f004:**
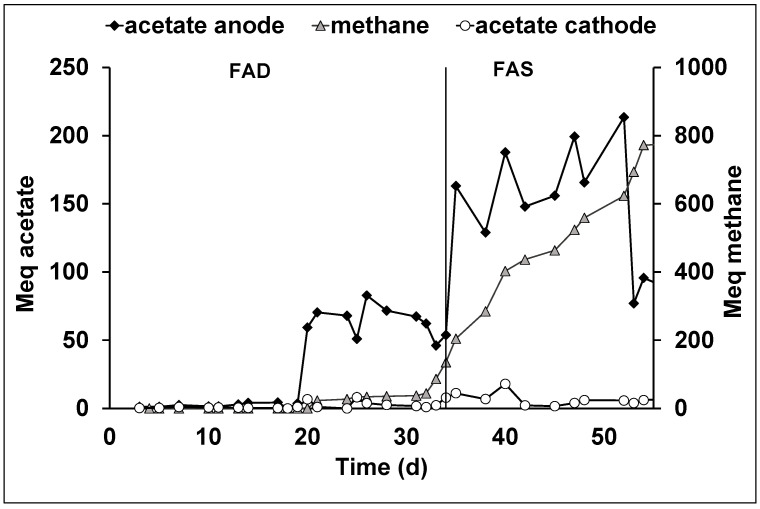
Time profile of the quantities of acetate and methane obtained in the continuous flow bioelectrochemical reactor inoculated with heat treated sludge in anodic and cathodic compartments.

**Figure 5 membranes-12-00126-f005:**
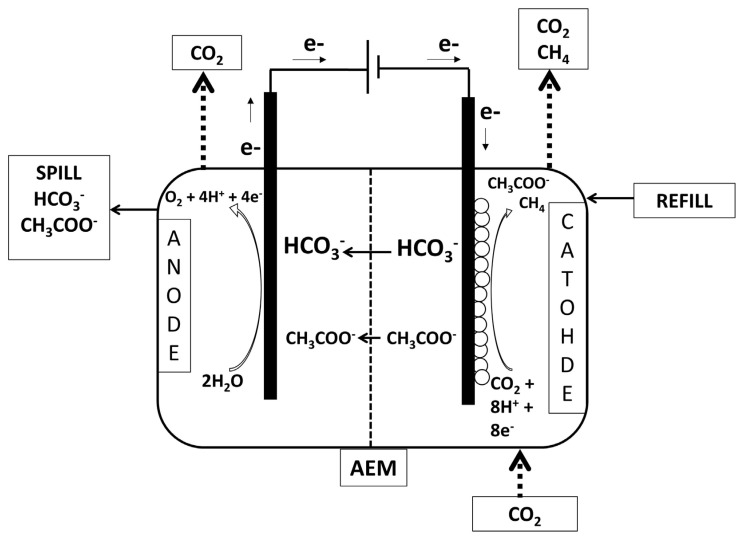
Schematic representation of the inorganic carbon mass balance for the continuous flow bioelectrochemical reactor.

**Figure 6 membranes-12-00126-f006:**
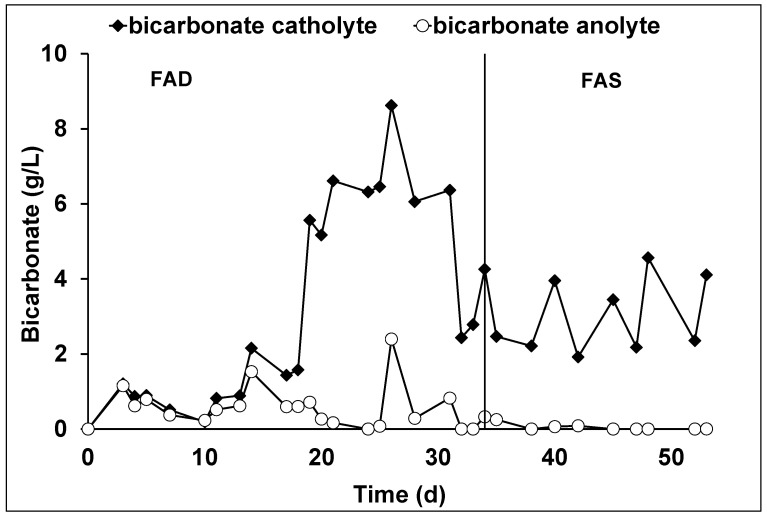
Time profile of the bicarbonate in the continuous flow bioelectrochemical reactor.

**Table 1 membranes-12-00126-t001:** Summary of main parameters representing the inorganic carbon mass balance.

mmol/d	AEM-FAD	AEM-FAS
CO_2_ removed	4.07 ± 0.76	7.75 ± 0.44
rCH_4_	0.70 ± 0.10	2.73 ± 0.30
rCH_3_COOH	0.07 ± 0.01	0.21 ± 0.07
CO_2_ out anode	0.02 ± 0.01	0.04 ± 0.01
Undefined	3.04 ± 0.04	4.85 ± 0.70

## Data Availability

Not applicable.

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
