# Peer review of "Autotrophic Acetate Production under Hydrogenophilic and Bioelectrochemical Conditions with a Thermally Treated Mixed Culture"

_membranes, 2022, doi:10.3390/membranes12020126_

Round 1

Reviewer 1 Report

The authors applied thermal treated digestate to enrich acetogenic inoculum and tested MEC-AEM reactor for selective acetate production. The manuscript was well written and it would induce interests of readers if it would be published in journal. The manuscript might be published to journal if the authors would be able to modify the manuscript in accordance with several comments below.
1)    Abstract should be enriched with important information that was derived from the study. Please reduce background and unnecessary information in Abstract section.
2)    It would be better to add reason why the authors selected AEM in Introduction section and capability of AEM to selectively transfer acetate to anode by referring previous studies.
3)    Please provide more detailed properties of FAS and FAD which showed different performances. 
4)    How did the authors believe that anaerobic digestate that the authors sampled can be acetogens source?. Moreover, How much acetogens did enrich in inoculum after thermal treatment ?. The authors should clarify these information because the authors did not analyze microbial community in digestate.
5)    Based on residual COD in digestate, other anion volatile fatty acids could be produced in cathodic chamber.

Author Response

Reviewer 1

The authors applied thermal treated digestate to enrich acetogenic inoculum and tested MEC-AEM reactor for selective acetate production. The manuscript was well written and it would induce interests of readers if it would be published in journal. The manuscript might be published to journal if the authors would be able to modify the manuscript in accordance with several comments below.

Abstract should be enriched with important information that was derived from the study. Please reduce background and unnecessary information in Abstract section.

Author: The abstract has been revised as suggested by the reviewer.

It would be better to add reason why the authors selected AEM in Introduction section and capability of AEM to selectively transfer acetate to anode by referring previous studies.

Author: A more in-depth explanation for the AEM utilisation for acetate recovery has been inserted in the introduction section (Line 78-85).

Please provide more detailed properties of FAS and FAD which showed different performances.

Author: Detailed proprieties of the FAS and FAD membrane have been included in the section 2.3 (Line 121-130) of the revised version of the manuscript.

How did the authors believe that anaerobic digestate that the authors sampled can be acetogens source? Moreover, how much acetogens did enrich in inoculum after thermal treatment? The authors should clarify this information because the authors did not analyse microbial community in digestate.

Author: Thanks for the request of clarification, the acetogens presence in the anaerobic digestate has been reported in several studies about the microbial community in the anaerobic digestate [i] before and after thermal treatment [ii]. Moreover, the presence of acetogens in anaerobic digestate is proved by the production of acetate in absence of organic substance as reported in the figure 2-A which shown a limited acetate production by the anaerobic digestate. The above-mentioned information has been inserted in the revised version of the manuscript in line 232-234.

[i]. [1] B.H. Yan, A. Selvam, S.Y. Xu, J.W.C. Wong, A novel way to utilize hydrogen and carbon dioxide in acidogenic reactor through homoacetogenesis, Bioresource Technology 159 (2014) 249-257.

[ii] M. Zeppilli, H. Chouchane, L. Scardigno, M. Mahjoubi, M. Gacitua, R. Askri, A. Cherif, M. Majone, Bioelectrochemical vs hydrogenophilic approach for CO2 reduction into methane and acetate, Chemical Engineering Journal. 396 (2020). https://doi.org/10.1016/j.cej.2020.125243.

Based on residual COD in digestate, other anion volatile fatty acids could be produced in cathodic chamber.

Author: Despite the possibility to convert COD in other VFA anions by an electrofermentation pathway [iii], no other volatile fatty acids other than acetate have been detected in all the reported bioelectrochemical tests. To increase the results discussion the above-mentioned sentence and the cited reference has been reported in the revised version of the manuscript (Line 279-282).

[iii] P. Paiano, M. Menini, M. Zeppilli, M. Majone, M. Villano, Electro-fermentation and redox mediators enhance glucose conversion into butyric acid with mixed microbial cultures, Bioelectrochemistry (2019) 107333.

Reviewer 2 Report

The researchers investigated autotrophic acetate production under hydrogenophilic and bioelectrochemical conditions. Language: please check your language style of your manuscript, especially in the introduction section. Please consider also to change the introduction by using a paragraph structure, thus it will be easier to understand the reasons behind conducting this study. In materials and methods what was in the mineral medium? In the Results section for the Hydrogenopilic batch tests was there a big difference between the result of the thermal treated inoculum (6 meq of acetate) in comparison with the control (5.25 meq of acetate)? Please discuss more on these results and it will help to add some references. Please change the x axis scale for the meq of acetate in Figure 2D from 0-15 to 0-9. It will be reasonable to add some references in page 6 line241, as also in general for the Bioelectrochemical batch-test results as there was no control or something else to compare those results. Also, in the Continuous bioelechtrochemical reactor section a comparison with previous studies is lacking i.e., a comparison of the acetate concentration produced could be added.

Author Response

Reviewer 2

The researchers investigated autotrophic acetate production under hydrogenophilic and bioelectrochemical conditions.

Language: please check your language style of your manuscript, especially in the introduction section. Please consider also to change the introduction by using a paragraph structure, thus it will be easier to understand the reasons behind conducting this study.

Author: the introduction as well as the language style has been revised according to the reviewer comment.

In materials and methods what was in the mineral medium?

Author: The composition of the mineral medium is reported in chapter 2.1 the section "inoculum's pre-treatment" (Line 90-92).

In the Results section for the Hydrogenophilic batch tests was there a big difference between the result of the thermal treated inoculum (6 meq of acetate) in comparison with the control (5.25 meq of acetate)? Please discuss more on these results and it will help to add some references.

Author: We thank the reviewer for the comment, maximum acetate concentration (meq) and acetate production rate (meq/Ld) have been revised in the revised version of the manuscript (section 3.1 Line 234-240).

Please change the x axis scale for the meq of acetate in Figure 2D from 0-15 to 0-9. It will be reasonable to add some references in page 6 line241, as also in general for the Bioelectrochemical batch-test results as there was no control or something else to compare those results.

Author: The axis value has been revised according to the reviewer comment. Moreover, as suggested by the reviewer, reference [24] been inserted also in line 260. Moreover, the bioelectrochemical batch test has been operated without a control because a previous study, which was conducted with similar condition showed the predominance of the methanogenesis reaction when a raw digestate was adopted as inoculum, indeed, as also discussed in line 232-234, the methanogenesis consortium showed a higher activity under bioelectrochemical conditions.

Also, in the Continuous bioelectrochemical reactor section a comparison with previous studies is lacking i.e., a comparison of the acetate concentration produced could be added.

Author: A comparison with a similar study has been added in section 3.3 line 319-322 by the insertion of the following sentence “Despite the accomplished acetate production and extraction, higher performances in terms of acetate production (0.7 g / Ld) and concentration (up to 35 g(/L) have been described in the literature thank to the utilization of specific antibiotics in the extraction chamber (anode compartment) and by the selection of an enriched acetogenic inoculum”.